# DATA-SCARCE DISTILLATION FOR LARGE-SCALE VISION LANGUAGE MODELS

## ABSTRACT

Vision-language models (VLMs) have emerged as extremely strong zero-shot and few-shot image classifiers, performing on par with task-specific models. However, they can be unnecessarily heavy-weight for task-specific downstream applications. While existing lines of work have successfully compressed VLMs and other foundation models to varying degrees, most focus on preserving the generality of these models, rather than leveraging their power for a particular task. In this work, we focus on the setting in which we have a limited amount of data on a downstream image classification task and a limited inference budget. To satisfy these constraints, we focus on distilling the strong few-shot performance of CLIP on image classification tasks into a more efficient model. We introduce the SID-CLIP (Synthesize-Initialize-Distill CLIP) method and highlight its three components that are critical to obtaining strong performance: 1) augmenting the classifier with *synthetic data* generated by leveraging CLIP itself; 2) *initializing* the modeling process using a smaller CLIP model pretrained on the target architecture; and 3) incorporating *knowledge distillation* to maximally mimic the performance of the larger model. Our set of proposed strategies produces a compact model that performs within 16% and 10% of CLIP's linear probe performance on 1 and 8 shot datasets respectively, while using a model with less than 2% of the parameters of CLIP's image encoder. We hope our work can be useful as a practical guide for leveraging the power of foundation models in downstream data-scarce and budget constrained settings.

## 1 INTRODUCTION

Foundation models such as CLIP-based models have been shown to perform extremely well on zero-shot and few-shot image classification: via simple prompting and/or a few examples, these models can achieve classification performance on-par with models trained with much more task-specific data (Radford et al., 2021). However, this performance comes at a cost: the models are extremely general and large-scale, and thus incur a high inference cost relative to smaller, more task-specific models, making them unsuitable for many edge applications. This challenge has led to a number of methods for compressing or distilling knowledge from large foundation models into smaller models. Although these techniques can preserve strong performance relative to the large foundation model, they are often not task-specific, and when they are, they often focus on preserving the model's zero-shot performance, rather than being able to take advantage of limited task-specific downstream data (Popp et al., 2024; Li et al., 2023; Wu et al., 2023; Vasu et al., 2024; Sun et al., 2023).

In this work, our goal is to produce a small model that performs as close as possible to a powerful large-scale vision-language model (VLM) on a particular downstream task. We address the specific challenge of attempting to distill the strong performance of zero- and few-shot CLIP image classification models into vastly more efficient (but task-specific) architectures. In other words, given a very limited amount of data on a desired downstream image classification task, and a very limited inference-time compute budget, we obtain the best performance on a downstream compact model by *leveraging the capabilities of larger models*. In practice, we find that three separate components are central to obtaining strong performance:

1. We augment the classifier with **synthetic data** generated by leveraging CLIP itself. Specifically, we use a text-to-image generative model seeded with embeddings produced from

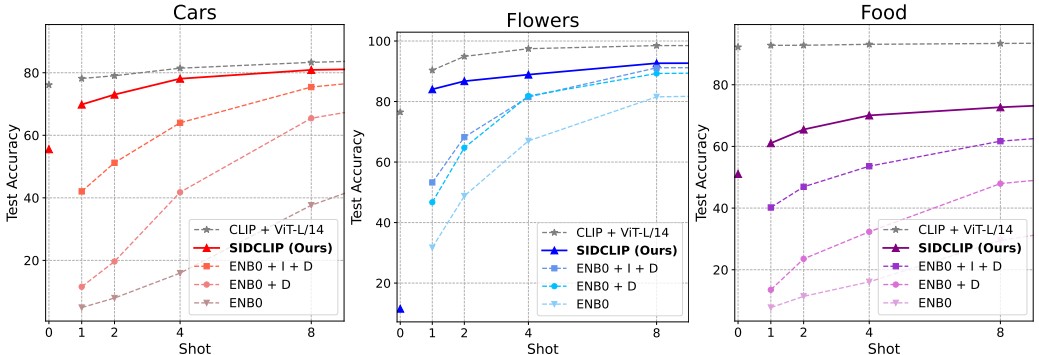

Figure 1: The addition of each SIDCLIP component increases performance, bringing the performance of the final model much closer to the performance of the teacher CLIP ViT-L/14 model, across shots and datasets.

> linear interpolations of the text of the class label *and* the CLIP embeddings of the few-shot image examples.

2. We **initialize** our small models with a variant based upon a small CLIP model pretrained on the target architecture.

3. We incorporate **knowledge distillation** to maximally mimic the performance of the larger CLIP models.

We call our method, which incorporates the above three components, *SIDCLIP* (Synthesize-Initialize-Distill CLIP). While each of these elements alone have been the subject of exploration in the literature, we emphasize that the work here serves largely as a "practical guide" that demonstrates the relative value of leveraging these three capabilities, as well as ablations demonstrating the efficacy of subsets of these elements.

We evaluate our proposed approaches, along with ablations and other baselines, on three common small-scale image classification benchmarks: the Stanford Cars (Krause et al., 2013), Oxford Flowers (Nilsback & Zisserman, 2008), and Food 101 (Bossard et al., 2014) datasets. We show that in all cases, our set of proposed strategies produces a compact model that performs within 16% and 10% of CLIP's linear probe performance on 1 and 8 shot datasets respectively, while having less than 2% of the parameters. These results are highlighted in Figure 1.

## 2 RELATED WORKS

**Synthetic data.** There has been lots of evidence to indicate that synthetic data is helpful in general when training models and particularly in distillation. Azizi et al. (2023) find that augmentation of a dataset with synthetic data improves image classification performance on CNN and ViT architectures. He et al. (2023) focuses on the zero- and few-shot domains and reaches a similar conclusion: that synthetic data can be used in conjunction with real data to improve performance on image classification tasks. Similarly to our work, Popp et al. (2024) generate synthetic data in order to perform distillation. This work differs from ours in two notable ways: they assume *no* access to the downstream data, rather than few samples; and the aim to transfer the general zero-shot capabilities of CLIP rather than focusing on a particular downstream task. In general, the field of data-free distillation explores the usage of only synthetic data (no real data) during the distillation process (Chawla et al., 2021; Fang et al., 2022).

While these directions all incorporate synthetic data, none utilize the particular image- and text-conditioning generation method that we use in SIDCLIP. The image generation pipeline we use was introduced in Razzhigaev et al. (2023) and achieved SOTA FID scores (a metric to assess the quality of generated images) on generated images relative to other open source models.

**Small CLIP models.** Several previous works attempt to compress the information in CLIP by reducing the size of the image encoder via distillation to a smaller CLIP model or to a novel CLIP-

like architecture (Wu et al., 2023; Vasu et al., 2024; Popp et al., 2024). In our work, we replace CLIP's image encoder with a new vision model, namely an EfficientNet B0 (Tan & Le, 2020).

**Compression.** VLMs have remarkable few- and zero-shot performance on downstream tasks and are strong image classifiers (Radford et al., 2021; Jia et al., 2021; Li et al., 2022; Yuan et al., 2021; Zhai et al., 2023). It is a natural next step to attempt to compress these high powered models into smaller versions that require less memory and have lower inference times. There has been a range of work in compressing foundation models, some (pruning, quantization, distillation) mirrors compression in non-foundation models, while others (parameter-efficient fine-tuning such as adapter layers or prompt tuning) are unique to the VLM or LLM setting (Hinton et al., 2015; Dettmers et al., 2022; Frantar & Alistarh, 2023; Sun et al., 2024; Houlsby et al., 2019; Liu et al., 2022; Lester et al., 2021; Jia et al., 2022).

In many of the existing efforts to compress foundation models, the goal has been to preserve the *general* capabilities of the models. Rather than honing in on a model's performance on a particular task, these methods aim to broadly preserve CLIP's generalizaiton abilities for image classification (Li et al., 2023; Wu et al., 2023; Vasu et al., 2024; Sun et al., 2023; Wu et al., 2022).

Li et al. (2023) distills from a CLIP ViT-L/14 teacher to a convolutional network student such as ResNet18. They measure task-specific performance as out-of-distribution performance: they perform distillation without any of the task-specific samples and then evaluate the zero- or few-shot performance of their model on downstream tasks. While similar to our setting, this setting does not take advantage of task-specific data during distillation and thus yields lower performance than our method. Although this allows for flexibility with downstream tasks, it is not the most advantageous when the downstream task is known ahead of time.

TinyCLIP and MobileCLIP both preserve CLIP's general purpose knowledge through distillation (Wu et al., 2023; Vasu et al., 2024). TinyViT is another method which produces a small downstream model via distillation (Wu et al., 2022). Task-specificity is not part of the distillation process for any of these methods.

Sun et al. (2023), like us, distill from CLIP ViT-L/14 to a smaller foundation model, and find that this distilled model outperforms a similar model trained from scratch. However, their smallest model (Swin-T) is over 5x larger than our model and they only report zero-shot numbers.

**Few-shot learning.** While preserving the entirety of CLIP's performance is a worthwhile goal, it is not the correct focus for all settings. The few-shot setting, when there is limited downstream training data available, arises in situations where data collection is expensive or challenging (Wang et al., 2020). Training large-scale models from scratch is an extremely data-intensive process, so usage of few-shot data to finetune an existing model can increase accessibility and customization of the power of models like VLMs. While there is some work that addresses a few-shot downstream setting, it is often done independently of the goal of compression, thus making these approaches not as feasible of solutions for resource-constrained users (Ma et al., 2024; Wortsman et al., 2022; Islam et al., 2021). Some lines of work, such as Popp et al. (2024), focus on the zero-shot setting, but leave out few-shot results. If some downstream task-specific training data is available, these methods are not equipped to best utilize it.

## 3 THE SIDCLIP METHOD

To use CLIP as an image classifier, first an image is passed into the image encoder, and text of the possible classnames is passed into the text encoder. Then, the embedding similarity between the image and possible classnames is measured. Although this process yields high accuracy on a variety of downstream tasks, the CLIP model is unnecessarily large and cumbersome for many downstream applications, such as use on edge devices. The best-performing and largest CLIP model, CLIP ViT-L/14, has 307M parameters in its image encoder Radford et al. (2021). But what if a user does not need the full "general-purpose" processing power of CLIP? They may want to take advantage of the off-the-shelf zero and few shot performance, but only need to classify images corresponding to a specific task and cannot afford to run such a large model. In this case, it is desirable to transfer a specific portion of CLIP's image classification capabilities to a smaller model.

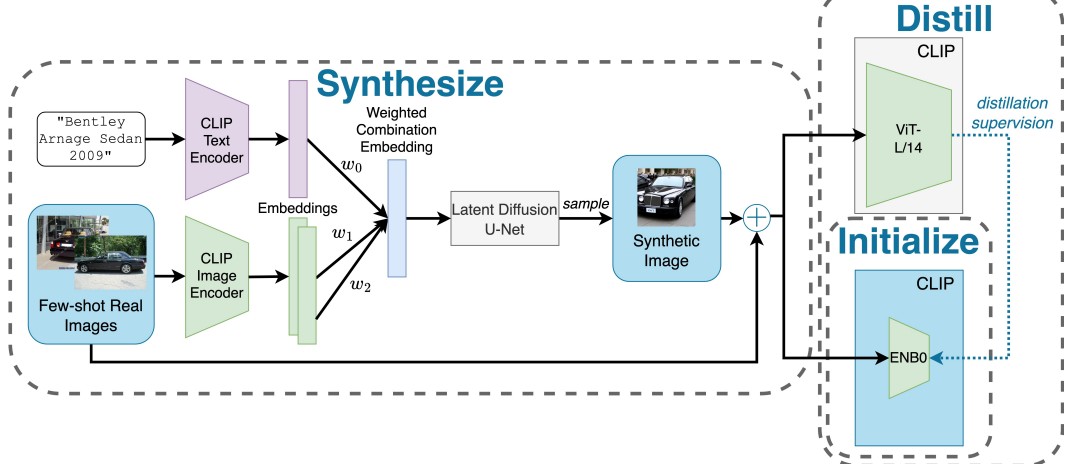

Figure 2: The three components of SIDCLIP: *synthesize* data via a weighted combination of class labels and real images; *initialize* the student as the image encoder of a small CLIP model; *distill* from a powerful teacher model.

With existing methods, a user would be able to produce a general-purpose small model, and potentially finetune it on the task of interest, but is left without being able to optimally take advantage of the limited training data they have. They would end up with a smaller version of CLIP, not a model tailored to their specific use case.

**Problem setting.** Suppose we are in the setting where we have access to a large-scale teacher VLM $\mathcal{T}$, such as CLIP. We have a small model architecture $\mathcal{S}$ that fits certain budget constraints. Our goal is to maximize image classification performance using $\mathcal{S}$ on some downstream task T. However, we only have $k$ labeled samples per class $c \in \mathcal{C}$, for $n = |\mathcal{C}|$ classes, on task T.

Our method, SIDCLIP, consists of three essential components for leveraging CLIP's power in training a small model in a data-constrained setting. These three components are 1) synthetic data, 2) initializing the model as a small CLIP variant, and 3) distilling from CLIP to the small model.

### 3.1 COMPONENT #1: SYNTHETIC DATA

We use synthetic data to augment the limited samples per class in a few shot setting. As described in our problem setting, we have $k$ labeled samples per class. We use these $k \times n$ samples $\mathcal{D}_r$ and their classname labels $\mathcal{L}$ to generate additional synthetic samples that can be used for training the model. When operating in the $k$-shot setting, we *only* use those $k$ samples and their classnames as input to generate additional synthetic data.

More formally, when we want to generate a synthetic sample from class $c$, we use the label $l_c \in \mathcal{L}$ and some set of images $\{x_i\}_{i=1}^{I} \in \mathcal{D}_{r,c}$, for $I \leq k$. We obtain the CLIP image and text embeddings: $\texttt{img\_enc}(x_i)$ and $\texttt{text\_enc}(l_c)$ and combine them via a weighted combination:

$$\texttt{emb} = w_0 \cdot \texttt{text\_enc}(l_c) + \sum_{i=1}^{I} w_i \cdot \texttt{img\_enc}(x_i)$$

such that $\sum_{i=0}^{I} w_i = 1$. This combination is then passed into the generative model, which we sample from to obtain synthetic samples $\{x_j'\} \sim \mathcal{G}(\texttt{emb})$, so that $\mathcal{D}_{s(r)} = \{x_j'\}_{j \in J}$.

In the majority of cases where synthetic data is used for training, images are generated based on solely a text prompt or a text prompt and an existing image (see Section 2). In this work, we aim to maximally leverage the existing data by utilizing a data generation pipeline which can take as input linear combinations of embeddings of text and *multiple* images.

Concretely, we use the Kandinsky framework, which takes as input real images and captions (Raz-zhigaev et al., 2023). This pipeline obtains CLIP embeddings for each image and caption, combines them according to specified weights, and passes the joint embedding into the diffusion model to produce a synthetic sample. We utilize this pipeline due to its high performance and flexibility: it achieved strong FID scores relative to competitors and was the first text-to-image generative model that used both image priors and latent diffusion.

## 3.2 COMPONENT #2: INITIALIZE AS SMALL CLIP

We find that initializing a student model in a CLIP-style architecture allows for performance gains relative to a standalone student vision model. In this paper, we distill to the EfficientNet B0 model, a small convolutional network with around 5.3M parameters (Tan & Le, 2020). For our primary set of experiments, we initialize this model as a small CLIP variant, that is preserving the CLIP text encoder and replacing the CLIP image encoder with an EfficientNet B0 model. This setup is pretrained on a subset of DataComp corresponding to 896M samples (Gadre et al., 2023).

## 3.3 COMPONENT #3: KNOWLEDGE DISTILLATION

Knowledge distillation is a common model compression technique that uses a large, powerful teacher model to train a smaller student model by aligning the student's output probabilities to those of the teacher. There are many variants of loss functions used to align these sets of probabilities, but the most common is based on the KL divergence as proposed in Hinton et al. (2015):

$$\mathcal{L}_{KL} = \alpha \cdot T^2 \cdot D_{\mathrm{KL}}(SM(\tilde{y}), SM(\hat{y})) + (1 - \alpha) \cdot CE(\hat{y}, y)$$

where $D_{\mathrm{KL}}$ refers to KL divergence, $CE$ refers to cross entropy, $SM$ refers to softmax, $\tilde{y}$ is the teacher output probabilities, $\hat{y}$ is the student output probabilities, $y$ is the true labels, $\alpha$ is a hyperparameter that trades off influence from teacher labels vs true labels, and $T$ is a temperature parameter.

We use this standard KL setting in our experiments. We have a teacher image encoder which outputs image embeddings of size $d_{img}^T$ and a student image encoder which outputs image embeddings of size $d_{img}^S$. We also have a common text encoder which produces text embeddings of size $d_{text}$.

For each task, we append a linear layer of shape $d_{img}^T \times c$ to the teacher image encoder and a similar layer of shape $d_{img}^S \times c$ to the student model. Before distillation, we finetune the teacher linear layer on the task of interest. We initialize the student layer with the text embeddings of each class: we obtain the embeddings for each caption `"A photo of {classname}."` or `"A photo of {classname}, a photo of {category}."` and concatenate them into a tensor of shape $d_{text} \times c$ where $d_{img}^S = d_{text}$. Then, during distillation, the teacher and its appended layer are frozen, and both the student and its appended layer are updated.

When performing distillation, we use a distillation set consisting of the real samples $\mathcal{D}_r$ and the synthetic samples $\mathcal{D}_{s(r)}$ generated using only those corresponding real images: $\mathcal{D} = \mathcal{D}_r \cup \mathcal{D}_{s(r)}$.

## 4 RESULTS

We demonstrate that SIDCLIP allows us to approach the performance of CLIP ViT-L/14 while using an image encoder with less than 2% of the parameters. Each of the three components (synthesize, initialize, distill) is critical in achieving this strong performance. Through a series of ablations and comparisons to SOTA distillation methods, we show that SIDCLIP is the dominant method when operating in a low inference budget, low data regime.

## 4.1 EXPERIMENTAL DETAILS

**Datasets.** We report results on three task-specific image classification datasets: StanfordCars, Ox-fordFlowers, and Food101 (Krause et al., 2013; Nilsback & Zisserman, 2008; Bossard et al., 2014). StanfordCars has 196 classes, OxfordFlowers has 102, and Food101 has 101. All numbers reported in the paper are top1 accuracy on the test sets.

Table 1: SIDCLIP outperforms competing methods and variants without all three components.

| Method | Params (M) | Shot 0 | 1 | 2 | 4 | 8 | Full |
|---|---|---|---|---|---|---|---|
| | | **Cars** | | | | | |
| CLIP ViT-L/14 | 307 | 76.1 | 78.17 | 79.04 | 81.46 | 83.32 | 91.1 |
| ENB0 | 5.3 | – | 4.34 | 7.0 | 15.92 | 37.74 | **87.79** |
| ENB0 + D | 5.3 | – | 11.5 | 19.64 | 39.45 | 59.22 | 85.03 |
| ENB0 + I + D | 5.3 | – | 42.06 | 51.22 | 64.0 | 75.43 | 86.58 |
| ENB0 + I + D + S (**SIDCLIP**) | 5.3 | **55.55** | **69.83** | **73.01** | **78.1** | **80.9** | 86.27 |
| TinyCLIP | 8 | 7.8 | 11.18 | 13.51 | 17.11 | 22.15 | 31.09 |
| TinyViT-5M | 5.4 | – | 3.05 | 5.83 | 13.9 | 29.2 | 87.65 |
| | | **Flowers** | | | | | |
| CLIP ViT-L/14 | 307 | 76.5 | 90.34 | 94.91 | 97.46 | 98.49 | 98.57 |
| ENB0 | 5.3 | – | 31.73 | 48.77 | 67 | 81.54 | 87.38 |
| ENB0 + D | 5.3 | – | 46.71 | 64.73 | 81.85 | 89.27 | 92.28 |
| ENB0 + I + D | 5.3 | – | 53.26 | 68.24 | 81.49 | 91.15 | 92.19 |
| ENB0 + I + D + S (**SIDCLIP**) | 5.3 | 11.53 | **84.04** | **86.73** | **88.89** | **92.65** | **93.35** |
| TinyCLIP | 8 | **56.46** | 69.25 | 78.19 | 86.75 | 90.21 | 82.44 |
| TinyViT-5M | 5.4 | – | 37.88 | 59.03 | 74.89 | 88.03 | 92.29 |
| | | **Food** | | | | | |
| CLIP ViT-L/14 | 307 | 92.2 | 92.78 | 92.8 | 93.11 | 93.39 | 95.15 |
| ENB0 | 5.3 | – | 7.73 | 11.37 | 16.08 | 29.47 | 83.5 |
| ENB0 + D | 5.3 | – | 13.5 | 23.54 | 32.31 | 47.94 | 80.13 |
| ENB0 + I + D | 5.3 | – | 40.19 | 46.9 | 53.58 | 61.7 | 87.25 |
| ENB0 + I + D + S (**SIDCLIP**) | 5.3 | 51.07 | **61.05** | **65.49** | **70.06** | **72.7** | **87.47** |
| TinyCLIP | 8 | **55.09** | 55.71 | 56.27 | 58.38 | 59.17 | 72.68 |
| TinyViT-5M | 5.4 | – | 9.38 | 16.67 | 21.0 | 33.58 | 84.65 |

**Data-scarce setting.** We are generally interested in any limited data setting. For experimental purposes, we simulate a data-scarce setting by creating few shot datasets from existing task-specific datasets. We randomly sample 1, 2, 4, or 8 images from each of the three datasets.

**Synthetic data.** We generate at least 300 synthetic images per class, per shot, and then sample from that pool to create our sets of 100, 200, and 300 shot synthetic samples. In all of the few-shot settings, our distillation dataset includes the few real samples per class and the synthetic samples generated from only those real samples. See Section A.1 for more details.

**Models.** We fix CLIP-ViT-L/14 as the teacher model (Radford et al., 2021). The student model is an EfficientNetB0 model initialized in a CLIP-style model (Tan & Le, 2020; ano, 2024). When performing distillation, the teacher model is frozen and we update the parameters of both the student model's image encoder and its appended linear layer. When performing finetuning of a CLIP-style model (CLIP ViT-L/14, TinyCLIP) we freeze the parameters of the student model and only update the parameters in the appended linear layer. When finetuning or distilling to a non-CLIP-style model (ENB0, ENB0 + D, TinyViT), there is no appended linear layer and we update all model parameters.

**Data augmentation.** We use RandAugment data augmentation (Cubuk et al., 2019). This is a strategy that applies random data augmentations to each image and is a top performing augmentation strategy. We apply six augmentations per image. See Section A.2 for additional discussion.

**Zero-shot results.** The zero-shot column always indicates that no real data was used. For our method (the SIDCLIP row), zero-shot distillation is performed by using 300 synthetic samples generated from only caption information. For the other rows (CLIP ViT-L/14, TinyCLIP), these numbers are text-conditioned evaluation on the test set. A dash indicates that zero-shot results were not obtained, either due to the model not being a CLIP-style model (ENB0, TinyViT-5M), or due to the inability to perform zero-shot distillation without synthetic data (ENB0 + D, ENB0 + I + D).

## 4.2 MAIN RESULTS

Table 1 shows a comparison of SIDCLIP to several relevant baselines. For each dataset, we include an "upper-bound" result: CLIP ViT-L/14, which is the teacher used in all experiments. We then include the baseline of a standalone EfficientNetB0 (ENB0) finetuned on the $k$-shot dataset. Each subsequent row adds one of the SIDCLIP elements: "+D" adds distillation, "+ I" adds initialization, and "+S" adds 300 synthetic samples in addition to the real samples.

We additionally include comparisons to two baseline methods: TinyCLIP and TinyViT (Wu et al., 2022; 2023). These methods use distillation to train a downstream image classification model. Unlike our method, which allows for specialization on a specific task, these methods focus on maintaining CLIP's overall performance. Additionally, few-shot results are not reported in these papers. For TinyCLIP, we ran few-shot linear probe experiments on the smallest available model (8M parameter image encoder). For TinyViT, we ran few-shot finetuning experiments on the smallest available model (5.4M parameters). Additional comparisons are included in section 4.3.

Our goal was to leverage the power of CLIP to produce a strong small-scale model, using only limited training data. These results indicate that, using each of our 3 components (synthesize, intialize, distill), we are able to obtain a notable performance increase of around +50% higher than the starting model, with performance that approaches that of the teacher CLIP model. Our method consistently outperforms variations which do not include all three components as well as few-shot finetuning of existing SOTA distillation methods.

On the Cars and Flowers datasets, SIDCLIP consistently achieves within 10% of CLIP's performance in the few shot setting. On Food, SIDCLIP remains 20-30% below the teacher model. We hypothesize that this may be due to more instances of food in the pretraining datasets for both teacher and student. In this case, additional food examples do not add much information to the model.

## 4.3 ADDITIONAL COMPARISONS

As previously discussed, many VLM compression methods focus only on the zero-shot or full-shot case. We include some additional results reported in the literature in Table 2. Our method compares favorably to the only other paper that reported few-shot results, with our 4-shot results outperforming their 5-shot results (Li et al., 2023). Although our method tends to perform worse than competitors on zero-shot, we note that the other models here are 2x larger, and our strong few-shot performance highlights the value of our data synthesis pipeline, which interpolates between real images and captions.

Table 2: Comparison to additional methods.

| Model | Params (M) | Zero shot | Few shot (k) | Full shot |
|---|---|---|---|---|
| **Cars** | | | | |
| TinyViT Popp et al. (2024) | 11 | **81.9** | – | **90.7** |
| ResNet18 Li et al. (2023) | 11 | 20.4 | 39.7 (5) | – |
| EfficientNet B0 (Ours) | 5.3 | 55.55 | **78.1** (4) | 86.27 |
| **Flowers** | | | | |
| TinyViT Popp et al. (2024) | 11 | **68.3** | – | 90.6 |
| ResNet18 Li et al. (2023) | 11 | 18.2 | 54.3 (5) | – |
| EfficientNet B0 (Ours) | 5.3 | 11.53 | **88.89** (4) | **92.29** |
| **Food** | | | | |
| TinyViT Popp et al. (2024) | 11 | **71.9** | – | 83.0 |
| ResNet18 Li et al. (2023) | 11 | 35.7 | 44.0 (5) | – |
| EfficientNet B0 (Ours) | 5.3 | 51.07 | **70.06** (4) | **87.47** |

## 4.4 ABLATIONS

**Synthetic data ablation** Our method uses 300 synthetic samples per class. In Figure 3 we include results with no added synthetic data or only 100 or 200 samples per class. There is a general trend

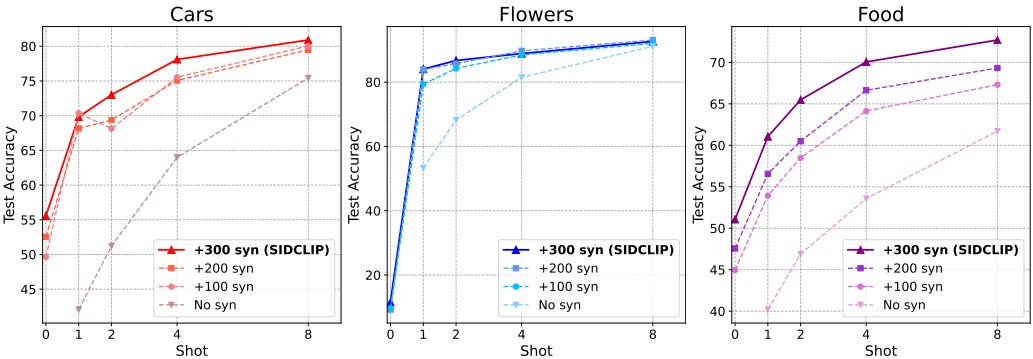

Figure 3: Ablation on amount of synthetic data.

of more synthetic data improving the performance, most notably in the smaller shot settings. However, the addition of the first 100 synthetic samples causes the largest increase in performance, with accuracy beginning to plateau with further addition of synthetic samples.

**Small CLIP ablation** Distilling to a vision model initialized in a small CLIP model results in higher performance than distilling to a standalone EfficientNet B0 model. This comparison is shown in Table 3. While our instantiation of the SIDCLIP method includes starting with a model that was pretrained on DataComp, our EfficientNet B0 model was pretrained on ImageNet. This discrepancy in pretraining dataset scale may also contribute to the difference in performance.

Table 3: Ablation of small CLIP initialization.

| Dataset | Method | Shot | | | | |
| | | 0 | 1 | 2 | 4 | 8 |
|---|---|---|---|---|---|---|
| Cars | ENB0 | 30.88 | 50.29 | 54.6 | 67.08 | 74.21 |
| | SIDCLIP (Ours) | **55.55** | **69.83** | **73.01** | **78.1** | **80.9** |
| Flowers | ENB0 | 3.59 | 83.02 | 85.25 | 89.74 | 91.77 |
| | SIDCLIP (Ours) | **11.53** | **84.04** | **86.73** | **88.89** | **92.65** |
| Food | ENB0 | 27.56 | 51.81 | 57.25 | 63.98 | 67.48 |
| | SIDCLIP (Ours) | **51.07** | **61.05** | **65.49** | **70.06** | **72.7** |

**Knowledge distillation ablation** In order to leverage the power of a large-scale VLM, we use knowledge distillation. Although knowledge distillation has long been established as a powerful compression technique (Hinton et al., 2015; Romero et al., 2014; Wu et al., 2023; 2022), here we include a simple ablation to demonstrate its value in our setting. We compare an EfficientNet B0 model (standalone; not initialized in a CLIP model) that is finetuned on a few shot dataset to one that is trained via distillation from a CLIP ViT-L/14 model. Table 4 shows that in all few-shot settings, distillation outperforms finetuning.

## 4.5 QUALITATIVE ANALYSIS OF SYNTHETIC IMAGES

Figure 4 shows examples of synthetic data used in the SIDCLIP pipeline. When conditioned on one or two real images as shown in the last two columns, we can see that the synthetic images directly mirror features in the real images more than when generation is only conditioned on the caption. For instance, note the colors of the Volkswagen Beetle and the butter on the waffles.

We also note in particular that the Flowers dataset tends to yield relatively poor zero-shot performance. We can observe how much the caption-only "red ginger" image differs from both the real images and the real-image-conditioned synthetic images. Additionally, the caption-only "yellow iris" includes less background foliage. This dataset-specific discrepancy may be a contributor to the impacted zero-shot performance.

Table 4: Ablation of distillation.

| Dataset | Method | Shot | | | |
|---|---|---|---|---|---|
| | | 1 | 2 | 4 | 8 |
| Cars | Finetune | 4.34 | 7.0 | 15.92 | 37.74 |
| | Distill | **11.5** | **19.64** | **39.45** | **59.22** |
| Flowers | Finetune | 31.73 | 48.77 | 67 | 81.54 |
| | Distill | **46.71** | **64.73** | **81.85** | **89.27** |
| Food | Finetune | 7.73 | 11.37 | 16.08 | 29.47 |
| | Distill | **13.5** | **23.54** | **32.31** | **47.94** |

| Caption | Real Image 1 | Real Image 2 | Synthetic image generation conditioned on: | | |
|---|---|---|---|---|---|
| | | | caption only | caption + real image 1 | caption + both real images |
| 'Rolls-Royce Phantom Sedan 2012' | | | | | |
| 'Volkswagen Beetle Hatchback 2012' | | | | | |
| 'red ginger' | | | | | |
| 'yellow iris' | | | | | |
| 'waffles' | | | | | |
| 'spaghetti bolognese' | | | | | |

Figure 4: Synthetic images mirror the real images more closely when conditioned on real images and captions, rather than captions only.

## 5 CONCLUSION

We present the SIDCLIP (Synthesize-Initialize-Distill CLIP) method, which achieves SOTA performance in a budget-constrained, data-scarce, task-specific setting. Our method achieves the best few-shot performance on StanfordCars, OxfordFlowers, and Food101 by leveraging the power of a large-scale VLM, in this instance, CLIP. The three components of the SIDCLIP method are 1) augmenting the limited training data with task-specific *synthetic data* generated by using linear combinations of the CLIP image and text embeddings of existing real data; 2) *initializing* the small model as a CLIP-style model; and 3) using *knowledge distillation* to transfer more fine-grained classification information from a powerful teacher. In settings with limited data and inference-time compute, SIDCLIP outperforms baselines such as TinyCLIP and TinyViT.

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

# A APPENDIX

## A.1 DETAILS OF SYNTHETIC IMAGE GENERATION

The caption we used to produce the text embedding is always the classname. For zero shot, we use only the caption to prompt the diffusion model and provide no real image samples. For 1 shot, we use the single image in each class as the only real image sample. For 2, 4, and 8 shot, we sample two images from each class of our few shot dataset. In the 1 shot case, we use weights of $0.4$ for text and $0.6$ for image, and for larger shots, we use weights of $0.2$ for the text and $0.4$ for each image.

## A.2 RANDAUGMENT DATA AUGMENTATION

We used RandAugment as the data augmentation method in our experiments. Here we ablate the usage of RandAugment, and show the results with only random flip and crop as data augmentations. Usage of RandAugment offers a performance increase, most notably in the smaller shot cases.

Table 5: Ablation of RandAugment

| Dataset | Method | Shot | | | | |
| | | 0 | 1 | 2 | 4 | 8 |
|---|---|---|---|---|---|---|
| Cars | No randaug | 50.37 | 61.82 | 63.9 | 71.56 | 77.32 |
| | SIDCLIP (Ours) | 55.55 | 69.83 | 73.01 | 78.1 | 80.9 |
| Flowers | No randaug | 8.46 | 79.04 | 81.31 | 86.42 | 90.88 |
| | SIDCLIP (Ours) | 11.53 | 84.04 | 86.73 | 88.89 | 92.65 |
| Food | No randaug | 41.1 | 52.4 | 56.83 | 64.24 | 67.54 |
| | SIDCLIP (Ours) | 51.07 | 61.05 | 65.49 | 70.06 | 72.7 |

## A.3 ABLATION ON AMOUNT OF SYNTHETIC DATA

Table 6: Ablation on amount of synthetic data.

| Dataset | Method | Shot | | | | |
| | | 0 | 1 | 2 | 4 | 8 |
|---|---|---|---|---|---|---|
| Cars | No syn data | − | 42.06 | 51.22 | 64 | 75.43 |
| | +100 syn | 49.65 | 70.35 | 68.14 | 75.54 | 80.05 |
| | +200 syn | 52.54 | 68.16 | 69.36 | 75.07 | 79.49 |
| | +300 syn (Ours) | 55.55 | 69.83 | 73.01 | 78.1 | 80.9 |
| Flowers | No syn data | − | 53.26 | 68.24 | 81.49 | 91.15 |
| | +100 syn | 9.66 | 79.36 | 84.26 | 88.45 | 92.05 |
| | +200 syn | 9.24 | 83.92 | 85.75 | 89.74 | 93.10 |
| | +300 syn (Ours) | 11.53 | 84.04 | 86.73 | 88.89 | 92.65 |
| Food | No syn data | − | 40.19 | 46.9 | 53.58 | 61.7 |
| | +100 syn | 44.95 | 53.93 | 58.49 | 64.14 | 67.31 |
| | +200 syn | 47.56 | 56.56 | 60.5 | 66.64 | 69.32 |
| | +300 syn (Ours) | 51.07 | 61.05 | 65.49 | 70.06 | 72.7 |

