# OpenReview forum: "Data-scarce distillation for large-scale vision language models"
_ICLR.cc/2025/Conference — Submitted to ICLR 2025_

### Official Review · Reviewer_QXk8 · 2024-10-28

**Soundness:** 3
**Presentation:** 3
**Contribution:** 3
**Rating:** 6
**Confidence:** 4

**Summary:**

This paper proposes a method to enhance model performance in data-scarce scenarios by distilling a smaller model using generated data derived from the CLIP model’s own image information. The paper also examines the impact of this generated data on model performance. While the approach shows some innovation in synthetic data generation, the overall methodology remains relatively simple and is validated only on three fine-grained datasets without experiments on larger-scale data.

**Strengths:**

1. The paper introduces a novel approach for synthetic data generation.
2. It improves edge deployment of smaller models by distilling knowledge from the CLIP model.

**Weaknesses:**

1. The experiments are limited in scale, with no evaluations on large datasets, such as ImageNet.
2. The range of compared models is narrow, including only EfficientNet-B0 (EB0).

**Questions:**

1. What would the performance be if the CLIP teacher model provided a logits-based distillation?
2. Does the proposed method perform similarly across all backbones? Specifically, how does it perform with larger models, such as ResNet-18 mentioned in the paper?
3. How is the quality of generated data ensured?
I would be glad to raise my score if the above questions could be addressed.

---

> ### Author Response · Authors · 2024-11-30
>
> We thank the reviewer for their thoughtful comments and willingness to reconsider their score.
>
> **Limited Experiments (W1)**
> > The experiments are limited in scale, with no evaluations on large datasets, such as ImageNet.
>
> Our focus here is on a particular task-specific setting. ImageNet includes a broad range of images but our work focuses on transferring a narrow set of knowledge from the teacher to the student. We consider settings where the user only needs their model to perform a limited, specific task (such as car type classification, not broad image categorization), and wants to leverage the power of foundation models to do so efficiently.
>
> **Additional Backbones (W2, Q2)**
> > The range of compared models is narrow, including only EfficientNet-B0 (EB0).
>
> >Does the proposed method perform similarly across all backbones? Specifically, how does it perform with larger models, such as ResNet-18 mentioned in the paper?
>
> We are unable to pretrain CLIP student models with a variety of backbones including ResNet but we include results for EfficientNet B1 and B2 models (with a more limited pretraining setup). We have added two additional backbones, EfficientNet B1 and EfficientNet B2. These CLIP-initialized models were pretrained with less DataComp data due to time constraints, so results are not directly comparable to EfficientNet B0 results, but we demonstrate that relative trends still hold. We include experiments on only the Stanford Cars dataset for the 1, 2, 4, and 8 shot settings. We note that SIDCLIP provides notable boosts relative to baselines for each model backbone.
>
> **8 shot**
> |                | EfficientNet B0 | EfficientNet B1 | EfficientNet B2 |
> |----------------|-----------------|-----------------|-----------------|
> | Baseline       | 37.74           | 45.18           | 49.2           |
> | +D             | 59.22           | 52.85           | 55.71          |
> | +I +D          | 75.43           | 70.34           | 69.64          |
> | SIDCLIP        | 80.9            | 74.37           | 74.95          |
>
> **4 shot**
> |                | EfficientNet B0 | EfficientNet B1 | EfficientNet B2 |
> |----------------|-----------------|-----------------|-----------------|
> | Baseline     | 15.92           | 21.0            | 20.84          |
> | +D             | 39.45           | 27.94           | 32.84          |
> | +I +D          | 64.0            | 55.35           | 55.09          |
> | SIDCLIP        | 78.1            | 68.13           | 68.37          |
>
> **2 shot**
> |                | EfficientNet B0 | EfficientNet B1 | EfficientNet B2 |
> |----------------|-----------------|-----------------|-----------------|
> | Baseline     | 7.0             | 8.67            | 8.72           |
> | +D             | 19.64           | 13.01           | 15.67          |
> | +I +D          | 51.22           | 35.75           | 34.71          |
> | SIDCLIP        | 73.01           | 58.46           | 60.3           |
>
> **1 shot**
> |                | EfficientNet B0 | EfficientNet B1 | EfficientNet B2 |
> |----------------|-----------------|-----------------|-----------------|
> | Baseline     | 4.34            | 5.02            | 4.78           |
> | +D             | 11.5            | 1.19            | 2.11           |
> | +I +D          | 42.06           | 23.5            | 23.88          |
> | SIDCLIP        | 69.83           | 63.56           | 64.35          |
>
> **Quality of Synthetic Data (Q3)**
> > How is the quality of generated data ensured?
>
> We would imagine an intuitive relationship between the quality of generated images and downstream results. We selected a performant existing diffusion model as the focus of our work is not on generative models. We did some limited hyperparameter tuning and manual evaluation of the generated images to ensure that fidelity was sufficient for the task at hand. However, we would like to emphasize that the added benefit of synthetic images is obtained without significant tuning effort needed for the generative model. Please note the images in Figure 4, which include representative samples from each dataset used in our paper. Our minimal hyperparameter tuning included selecting the weighting between images and text, and selecting to perform an interpolation between two images and one text prompt.

---

> > ### Comment · Reviewer_QXk8 · 2024-12-03
> >
> > Thanks for authors' efforts.
> > 1. I think this study could potentially benefit from the integration of more sophisticated data generation techniques, utilizing advanced generation models such as Flux, image filtering, image-text similarity assessments, and multimodal models, which could significantly enhance performance.
> > 2. I continue to emphasize the necessity for verification of this work on a large-scale dataset to ensure its robustness and applicability.
> > 3. The authors have conducted additional experiments using various backbone architectures to substantiate the effectiveness of their method.
> > Overall, the authors have addressed several pertinent questions. So, I have revised my score from 5 to 6.

---

### Official Review · Reviewer_w72x · 2024-11-02

**Soundness:** 2
**Presentation:** 2
**Contribution:** 2
**Rating:** 3
**Confidence:** 4

**Summary:**

This paper proposes a framework to distill a vision-language model (CLIP) in a data-scarcity setting. Starting from a CLIP-ViT-L/14 pretrained model and a few-shot downstream data of interest, the authors propose to train a small network (EfficientNet B0) in order to tailor it for the specific dataset at hand. The framework, called SIDCLIP, consists of three components: 1- Synthesizing data using a pretrained latent diffusion model that takes as input a CLIP embedding, namely a linear combination of the text embedding describing the class and embeddings of images from the same class coming from the few-shot dataset, 2- Initializing the small student network by CLIP-like pretraining on DataComp, and 3- knowledge distillation from the large teacher model (i.e. CLIP-ViT-L/14) using classical logit matching. Experiments on 3 datasets demonstrate the effectiveness of each of the 3 components, as well as advantages compared to other "small" general-purpose baselines in the few-shot setting.

**Strengths:**

* The paper is written in a simple language and is easy to understand and follow.
* The combination of synthetic data generated using the few-shot samples, initialization with CLIP-like pretraining, and knowledge distillation shows a good performance on the downstream dataset, and ablation shows the effectiveness of each component.

**Weaknesses:**

* Besides combining synthetic data, initialization and distillation for the data-scarcity specific setting, the proposed approach is limited in novelty. The main elements (distillation [1] and data generation [2]) are straightforwardly utilized. While their combination is interesting, the results are not very surprising due to the expected complementary knowledge coming from different sources of information: the generation capacity of latent diffusion informed with the specific dataset at hand, the pretrained ViT-L/14 knowledge and the DataComp data for pretraining as initialization.
* While the classical logit-based knowledge distillation approach seems to be effective, more advanced techniques were introduced in the literature ([A],[B],[C]). Since the approach is straightforward, it would be interesting to explore some other distillation methods as a positioning among other design choices. For example, some distillation methods were shown to improve classical distillation by decoupling target class and non-target class terms [B]. Other methods  show that distillation is also possible for "smaller" and "weaker" teacher [A]. Investigating such choices might help answering the question whether ViT-L/14 is needed as a teacher, or if the same or close results can be achieved with a smaller version of CLIP as teacher.
* The method is tested on only 3 datasets. Corroboration on more datasets helps assessing the value of the proposed framework and its practical use, especially that the target is a data-specialized model not a general purpose one. For example, the authors could add more experiments on common datasets used for few-shot learning with CLIP: ImageNet, DTD, EuroSAT, FGVCAircraft, SUN397, Caltech101... Please refer to [D] for more examples of such datasets. Some of these dataset (like DTD, EuroSAT, FGVCAircraft) differ from the used ones in being more specialized and fine-grained. How would each of the components help the total performance for such datasets?
* Compared to the baselines (TinyCLIP,TinyViT-5M), SIDCLIP uses a latent diffusion model, which makes the comparison unfair, and even the data-scarcity setting no more holding. Moreover, using DataComp pretraining data for the initialization stage makes the comparison more difficult, as TinyCLIP and TinyViT do not use the same dataset.

[1] Hinton et al., Distilling the Knowledge in a Neural Network. NeurIPS Workshops 2014
[2] Razzhigaev et al., Kandinsky: an Improved Text-to-Image Synthesis with Image Prior and Latent Diffusion. arxiv 2023
[A] Yuan et al., Revisiting knowledge distillation via label smoothing regularization. CVPR 2020
[B] Zhao et al., Decoupled Knowledge Distillation. CVPR 2022
[C] Beyer et al., Knowledge distillation: A good teacher is patient and consistent. CVPR 2022
[D] Zhou et al., Learning to Prompt for Vision-Language Models. IJCV 2022

**Questions:**

* For the ablation in Figure 1, are there any specific reasons for not showing other options like ENB0+S or ENB0+S+I? For example training ENB0+S (using synthetic data, no specific initialization, no distillation), might help better assessing the role of synthetic data.
* It is not clear how the linear probe (LP) was trained. Properly training LP is detrimental for getting the best of it, and classification results might largely fluctuate depending on the fine-tuning strategy [E],[F]. Could the authors give more details on this please?

[E] Yu et al., Task Residual for Tuning Vision-Language Models. CVPR 2023
[F] Huang et al., LP++: A Surprisingly Strong Linear Probe for Few-Shot CLIP. CVPR 2024

---

> ### Author Response · Authors · 2024-11-30
>
> We would like to thank the reviewer for their thorough feedback.
>
> **Limited Novelty (W1)**
> > Besides combining synthetic data, initialization and distillation for the data-scarcity specific setting, the proposed approach is limited in novelty. The main elements (distillation [1] and data generation [2]) are straightforwardly utilized. While their combination is interesting, the results are not very surprising due to the expected complementary knowledge coming from different sources of information: the generation capacity of latent diffusion informed with the specific dataset at hand, the pretrained ViT-L/14 knowledge and the DataComp data for pretraining as initialization.
>
> We agree that our work is primarily a practical guide to knowledge distillation that leverages foundation models and that the combination of our methods may not be surprising. However, we believe that despite the simplicity of our approach, it still carries value. Synthetic data does not always improve performance, and has been shown to cause model collapse in some settings. Distillation can also be challenging across large size gaps between the teacher and student, particularly across different architectures. Although our result may not be surprising, we believe it may be valuable to the community to demonstrate the combined benefit of our approaches, whether it provides a baseline for future research into foundation model distillation, or whether it can be used as a guide for users who simply want a small but performant task-specific model.
>
> **Additional Distillation Methods (W2)**
> > While the classical logit-based knowledge distillation approach seems to be effective, more advanced techniques were introduced in the literature ([A],[B],[C]). Since the approach is straightforward, it would be interesting to explore some other distillation methods as a positioning among other design choices. For example, some distillation methods were shown to improve classical distillation by decoupling target class and non-target class terms [B]. Other methods show that distillation is also possible for "smaller" and "weaker" teacher [A]. Investigating such choices might help answering the question whether ViT-L/14 is needed as a teacher, or if the same or close results can be achieved with a smaller version of CLIP as teacher.
>
> We agree that fruitful future work could examine the effect of using a smaller or weaker teacher model. However, in our paper, we choose to limit ourselves to the setting where we have access to a strong teacher model. Of the methods listed here, we we able to include additional results on decoupled knowledge distillation [B]. We include a limited set of experiments here where we use the CLIP-initialized EfficientNet B0 and the StanfordCars 8 shot dataset. SIDCLIP achieves the highest performance when using DKD. We use alpha=1 and beta=8. We would imagine that other distillation methods would hold a similar pattern, and we will include these results in our final paper.
>
> |                | EfficientNet B0 |
> |----------------|-----------------|
> | Baseline       | 37.74           |
> | +D             | 36.95           |
> | +I +D          | 58.7            |
> | SIDCLIP        | 77.02           |
>
> **Limited Datasets (W3)**
> > The method is tested on only 3 datasets. Corroboration on more datasets helps assessing the value of the proposed framework and its practical use, especially that the target is a data-specialized model not a general purpose one. For example, the authors could add more experiments on common datasets used for few-shot learning with CLIP: ImageNet, DTD, EuroSAT, FGVCAircraft, SUN397, Caltech101... Please refer to [D] for more examples of such datasets. Some of these dataset (like DTD, EuroSAT, FGVCAircraft) differ from the used ones in being more specialized and fine-grained. How would each of the components help the total performance for such datasets?
>
> We agree that more datasets would help paint a more comprehensive picture of our results, but unfortunately we are limited by computational constraints. We chose our three datasets for their specialized task-specificity. Each of the datasets we chose is limited to a narrow domain (cars, flowers, food) and the classes within each dataset highlight fine-grained differences between otherwise somewhat similar images.

---

> > ### Author Response · Authors · 2024-11-30
> >
> > **Fair Comparisons (W4)**
> > > Compared to the baselines (TinyCLIP,TinyViT-5M), SIDCLIP uses a latent diffusion model, which makes the comparison unfair, and even the data-scarcity setting no more holding. Moreover, using DataComp pretraining data for the initialization stage makes the comparison more difficult, as TinyCLIP and TinyViT do not use the same dataset.
> >
> > We would argue that our method is still in a data-scarce setting, since we have limited real data. Our work aims to highlight how, if a user has limited real data but access to resources to generate synthetic images, SIDCLIP can add notable value. In many settings, collection or labeling of real images can be very costly relative to the price of synthetic data generation and our work aims to provide value in these settings. We aim to provide a straightforward comparison between our method and existing other methods, but we could imagine interesting future work exploring the addition of synthetic data to the distillation processes of TinyCLIP and TinyViT. With regards to the DataComp pretraining data, we agree that the process does not exactly parallel that of TinyCLIP and TinyViT. However, our goal was to produce an end-to-end pipeline that can be useful in certain data-scarce settings. We believe that our method offers a unique advantage in these situations and while we demonstrate comparisons to similar existing baselines, these baselines do not aim to operate in the exact same types of situations and therefore provide an imperfect comparison.
> >
> > **Additional Ablations (Q1)**
> > > For the ablation in Figure 1, are there any specific reasons for not showing other options like ENB0+S or ENB0+S+I? For example training ENB0+S (using synthetic data, no specific initialization, no distillation), might help better assessing the role of synthetic data.
> >
> > We agree that comprehensive ablations paint a better picture of the strength of our model. Here we include a comprehensive set of ablations of each set of components for a subset of our experimental settings, namely 2, 4, and 8 shot versions of StanfordCars using the EfficientNet B0 model. We will add these to the final paper. We note that synthetic data can sometimes detract from overall performance, but in conjunction with distillation and initialization, it consistently outperforms baselines.
> >
> > |               | 2 Shot | 4 Shot | 8 Shot |
> > |---------------|--------|--------|--------|
> > | ENB0          | 7.0    | 15.92  | 37.74  |
> > | ENB0 + I      | 64.66  | 68.88  | 73.34  |
> > | ENB0 + D      | 19.64  | 39.45  | 59.22  |
> > | ENB0 + S      | 18.95  | 35.33  | 53.13  |
> > | ENB0 + I + D  | 51.22  | 64.0   | 75.43  |
> > | ENB0 + I + S  | 28.73  | 26.13  | 32.21  |
> > | ENB0 + S + D  | 54.6   | 67.08  | 74.21  |
> > | SIDCLIP       | 73.01  | 78.1   | 80.9   |
> >
> >
> > **Linear Probe Training (Q2)**
> > >It is not clear how the linear probe (LP) was trained. Properly training LP is detrimental for getting the best of it, and classification results might largely fluctuate depending on the fine-tuning strategy [E],[F]. Could the authors give more details on this please?
> >
> > For linear probe experiments, we appended a single linear layer of shape output_dim x num_classes to the model. Then to train this layer, we freeze the weights in the model and only train the weights in the linear layer.

---

### Official Review · Reviewer_MCuE · 2024-11-03

**Soundness:** 4
**Presentation:** 4
**Contribution:** 3
**Rating:** 6
**Confidence:** 3

**Summary:**

The paper focuses on the issues of limited data for a fine-grained downstream task and limited inference budget, and introduces a method called (Synthesize-Initialize-Distill CLIP) (SIDCLIP), involving three components:
- Synthesize (S): a novel approach to generate synthetic data by leveraging CLIP;
- Initialize (I): model initialization of a smaller CLIP model pretrained on the target architecture;
- Distill (D): knowledge distillation to a larger model.
Experimental results are reported on three task-specific image classification datasets: StanfordCars, OxfordFlowers, and Food101.

**Strengths:**

The author(s) show a lot of analysis for experiments to support their claim. While the proposal involves three components, there are specific ablation studies to show the advantage of the new model design and the comparison of eval metrics and visualization of synthetic image to exhibit the data quality.

**Weaknesses:**

The new proposal is extending several existing methods and putting things together to reach some better performance. While this is innovative, it will be more convincing to analyze the role of each component for the overall performance improvement.

Line 339 has a typo "intialize."

**Questions:**

Related to the weakness, for example, will the synthetic data help other models also?

---

> ### Author Response · Authors · 2024-11-30
>
> Thank you to the reviewer for their generally positive comments and rating.
>
> **Ablations (W1)**
> > The new proposal is extending several existing methods and putting things together to reach some better performance. While this is innovative, it will be more convincing to analyze the role of each component for the overall performance improvement.
>
> We agree that comprehensive ablations paint a better picture of the strength of our model. Here we include a comprehensive set of ablations of each set of components for a subset of our experimental settings, namely 2, 4, and 8 shot versions of StanfordCars using the EfficientNet B0 model. We will add these to the final paper. We note that synthetic data can sometimes detract from overall performance, but in conjunction with distillation and initialization, it consistently outperforms baselines.
>
>
> |               | 2 Shot | 4 Shot | 8 Shot |
> |---------------|--------|--------|--------|
> | ENB0          | 7.0    | 15.92  | 37.74  |
> | ENB0 + I      | 64.66  | 68.88  | 73.34  |
> | ENB0 + D      | 19.64  | 39.45  | 59.22  |
> | ENB0 + S      | 18.95  | 35.33  | 53.13  |
> | ENB0 + I + D  | 51.22  | 64.0   | 75.43  |
> | ENB0 + I + S  | 28.73  | 26.13  | 32.21  |
> | ENB0 + S + D  | 54.6   | 67.08  | 74.21  |
> | SIDCLIP       | 73.01  | 78.1   | 80.9   |
>
>
> **Additional Models (Q1)**
> > Related to the weakness, for example, will the synthetic data help other models also?
>
> We have added two additional backbones, EfficientNet B1 and EfficientNet B2. These CLIP-initialized models were pretrained with less DataComp data due to time constraints, so results are not directly comparable to EfficientNet B0 results, but we demonstrate that relative trends still hold. We include experiments on only the Stanford Cars dataset for the 1, 2, 4, and 8 shot settings. We note that SIDCLIP provides notable boosts relative to baselines for each model backbone.
>
> **8 shot**
> |                | EfficientNet B0 | EfficientNet B1 | EfficientNet B2 |
> |----------------|-----------------|-----------------|-----------------|
> | Baseline       | 37.74           | 45.18           | 49.2           |
> | +D             | 59.22           | 52.85           | 55.71          |
> | +I +D          | 75.43           | 70.34           | 69.64          |
> | SIDCLIP        | 80.9            | 74.37           | 74.95          |
>
> **4 shot**
> |                | EfficientNet B0 | EfficientNet B1 | EfficientNet B2 |
> |----------------|-----------------|-----------------|-----------------|
> | Baseline     | 15.92           | 21.0            | 20.84          |
> | +D             | 39.45           | 27.94           | 32.84          |
> | +I +D          | 64.0            | 55.35           | 55.09          |
> | SIDCLIP        | 78.1            | 68.13           | 68.37          |
>
> **2 shot**
> |                | EfficientNet B0 | EfficientNet B1 | EfficientNet B2 |
> |----------------|-----------------|-----------------|-----------------|
> | Baseline     | 7.0             | 8.67            | 8.72           |
> | +D             | 19.64           | 13.01           | 15.67          |
> | +I +D          | 51.22           | 35.75           | 34.71          |
> | SIDCLIP        | 73.01           | 58.46           | 60.3           |
>
> **1 shot**
> |                | EfficientNet B0 | EfficientNet B1 | EfficientNet B2 |
> |----------------|-----------------|-----------------|-----------------|
> | Baseline     | 4.34            | 5.02            | 4.78           |
> | +D             | 11.5            | 1.19            | 2.11           |
> | +I +D          | 42.06           | 23.5            | 23.88          |
> | SIDCLIP        | 69.83           | 63.56           | 64.35          |
>
> **Typos (W2)**
> Thank you for pointing this out - we will fix this typo and double check our paper for copy editing errors.

---

> ### Comment · Reviewer_MCuE · 2024-12-03
>
> Thanks for the additional experiments, which make the work more solid for my previous 6 rating. Yet, I do not plan to increase it further.

---

### Official Review · Reviewer_nfdE · 2024-11-05

**Soundness:** 3
**Presentation:** 3
**Contribution:** 2
**Rating:** 5
**Confidence:** 3

**Summary:**

This paper proposed SIDCLIP: Synthesize-Initialize-Distill CLIP, for few-shot distillation in data-scarce setting. SIDCLIP first augments limited labeled data using text-to-image generation model. Then, it trains a small model initialized with small CLIP. Finally it transfers knowledge from a large CLIP to the smaller model. Experiments on Stanford Cars, Oxford Flowers, Food-101 demonstrates the effectiveness of the proposed approach.

**Strengths:**

* Unlike general-purpose distillation methods, the proposed approach focuses on specific downstream tasks, being highly effective in specialized applications.

* SIDCLIP has a fraction of the parameters of the original CLIP, making it computationally and parameter efficient.

**Weaknesses:**

* The use of text-to-image generation to create synthetic data can introduce computational overhead, potentially making the framework slower or more resource-intensive.

* The paper primarily experiments with EfficientNet B0 as the small model architecture. How does SIDCLIP perform with different or larger backbones? Testing with a broader range of architectures would help demonstrating the generalizability of SIDCLIP.

* While SIDCLIP focuses on task-specific performance, its robustness in handling out-of-domain data (datasets significantly different from those in CLIP’s pretraining) is questionable. Since most of CLIP’s parameters remain frozen, there may be limitations in SIDCLIP’s adaptability to new domains, which could restrict its effectiveness in real-world, diverse data settings.

* The combination of synthetic data generation, initialization, and distillation steps may limit SIDCLIP’s practicality in scenarios requiring minimal preprocessing. The entire pipeline involves multiple stages that could increase setup and training time. It would be beneficial to provide the computation requirements compared to existing methods.

**Questions:**

* Could the authors provide more quantitative details on SIDCLIP’s training efficiency compared to other distillation methods?

* How sensitive is SIDCLIP’s performance to the quality of synthetic data generated? How would different generation models (simpler generative model instead of Kandinsky) affect the few-shot results?

---

> ### Author Response · Authors · 2024-11-30
>
> We would like to thank the reviewer for their thoughtful comments.
>
>
> **Computational requirements (W1, W4, Q1)**
> > The use of text-to-image generation to create synthetic data can introduce computational overhead, potentially making the framework slower or more resource-intensive.
>
> > The combination of synthetic data generation, initialization, and distillation steps may limit SIDCLIP’s practicality in scenarios requiring minimal preprocessing. The entire pipeline involves multiple stages that could increase setup and training time. It would be beneficial to provide the computation requirements compared to existing methods.'
>
> > Could the authors provide more quantitative details on SIDCLIP’s training efficiency compared to other distillation methods?
>
> The data generation process can be expensive. This work focuses on a setting in which a user has limited real data and wants to have low inference latency. Therefore, we assume that some upfront cost is acceptable. This upfront, one-time cost can be viewed as being amortized over an improved inference time. The user spends some cost to generate synthetic data and perform distillation, but at the end of this process, the resulting small model has high performance and low inference latency and can be used with less overhead than other alternatives, such as using the teacher at inference time. We provide some of the times required for these processes. Using one Nvidia A6000 GPU, the generation of 100 images takes 840.6 seconds. The difference in time for one epoch when using just real data vs real + synthetic data can primarily be ascribed to the increased amount of data being processed. For the 8 real shot setting (8 images per class), our CLIP-initialized EfficientNet B0 model takes around 24 seconds for one epoch of distillation, and the 8 real + 300 syn shot setting (308 total images per class) takes around 1021 seconds for one epoch of distillation.
>
>
> **Additional Backbones (W2)**
> > The paper primarily experiments with EfficientNet B0 as the small model architecture. How does SIDCLIP perform with different or larger backbones? Testing with a broader range of architectures would help demonstrating the generalizability of SIDCLIP.
>
> We have added two additional backbones, EfficientNet B1 and EfficientNet B2. These CLIP-initialized models were pretrained with less DataComp data due to time constraints, so results are not directly comparable to EfficientNet B0 results, but we demonstrate that relative trends still hold. We include experiments on only the Stanford Cars dataset for the 1, 2, 4, and 8 shot settings. We note that SIDCLIP provides notable boosts relative to baselines for each model backbone.
>
> **8 shot**
> |                | EfficientNet B0 | EfficientNet B1 | EfficientNet B2 |
> |----------------|-----------------|-----------------|-----------------|
> | Baseline       | 37.74           | 45.18           | 49.2           |
> | +D             | 59.22           | 52.85           | 55.71          |
> | +I +D          | 75.43           | 70.34           | 69.64          |
> | SIDCLIP        | 80.9            | 74.37           | 74.95          |
>
> **4 shot**
> |                | EfficientNet B0 | EfficientNet B1 | EfficientNet B2 |
> |----------------|-----------------|-----------------|-----------------|
> | Baseline     | 15.92           | 21.0            | 20.84          |
> | +D             | 39.45           | 27.94           | 32.84          |
> | +I +D          | 64.0            | 55.35           | 55.09          |
> | SIDCLIP        | 78.1            | 68.13           | 68.37          |
>
> **2 shot**
> |                | EfficientNet B0 | EfficientNet B1 | EfficientNet B2 |
> |----------------|-----------------|-----------------|-----------------|
> | Baseline     | 7.0             | 8.67            | 8.72           |
> | +D             | 19.64           | 13.01           | 15.67          |
> | +I +D          | 51.22           | 35.75           | 34.71          |
> | SIDCLIP        | 73.01           | 58.46           | 60.3           |
>
> **1 shot**
> |                | EfficientNet B0 | EfficientNet B1 | EfficientNet B2 |
> |----------------|-----------------|-----------------|-----------------|
> | Baseline     | 4.34            | 5.02            | 4.78           |
> | +D             | 11.5            | 1.19            | 2.11           |
> | +I +D          | 42.06           | 23.5            | 23.88          |
> | SIDCLIP        | 69.83           | 63.56           | 64.35          |

---

> > ### Author Response · Authors · 2024-11-30
> >
> > **OOD Robustness (W3)**
> > > While SIDCLIP focuses on task-specific performance, its robustness in handling out-of-domain data (datasets significantly different from those in CLIP’s pretraining) is questionable. Since most of CLIP’s parameters remain frozen, there may be limitations in SIDCLIP’s adaptability to new domains, which could restrict its effectiveness in real-world, diverse data settings.
> >
> > The goal in this work is task specificity. Our aim is to leverage a foundation model to perform a specific downstream task in a resource constrained setting. Robustness strategies are compatible with our approach (i.e. we use RandAugment data augmentation and we imagine that other augmentation or robust training approaches could be integrated into our pipeline) but analyzing the robustness to out of distribution images is not the primary focus of our work.
> >
> > **Synthetic Data Quality (Q2)**
> > > How sensitive is SIDCLIP’s performance to the quality of synthetic data generated? How would different generation models (simpler generative model instead of Kandinsky) affect the few-shot results?
> >
> > We would imagine an intuitive relationship between the quality of generated images and downstream results. We selected a performant existing diffusion model as the focus of our work is not on generative models. We did some limited hyperparameter tuning and manual evaluation of the generated images to ensure that fidelity was sufficient for the task at hand. However, we would like to emphasize that the added benefit of synthetic images is obtained without significant tuning effort needed for the generative model. Please note the images in Figure 4, which include representative samples from each dataset used in our paper. Our minimal hyperparameter tuning included selecting the weighting between images and text, and selecting to perform an interpolation between two images and one text prompt.

---

### Meta-Review · Area_Chair_UJao · 2024-12-20

**Metareview:**

The paper introduces a novel method for data-scarce distillation in large-scale vision-language models, focusing on leveraging CLIP's few-shot performance for efficient model deployment in resource-constrained settings. The key contribution is the introduction of the Synthesize-Initialize-Distill CLIP method, which combines synthetic data generation, model initialization, and knowledge distillation to produce a compact model while maintaining high performance.

However, there are concerns regarding the computational overhead introduced by synthetic data generation and its potential impact on the efficiency of the framework. Additionally, the novelty of combining existing methods, such as distillation and synthetic data generation, and the generalizability of the approach across different architectures and datasets were questioned. The reviewers' feedback suggests that a deeper analysis of the contribution of each component to the overall performance improvement would strengthen the paper.

Despite receiving final scores of 3, 5, 6, 6, and an average score of 5.0, the AC notes that there is significant potential in the proposed method but suggests that the authors revise the paper by addressing the reviewers' concerns, particularly regarding the novelty and generalization of the proposed solution. The AC encourages the authors to consider resubmitting the paper to the next conference with these revisions.

**Additional Comments On Reviewer Discussion:**

The reviewers raised concerns regarding the computational overhead introduced by synthetic data generation and its potential impact on the efficiency of the framework. Additionally, the novelty of combining existing methods, such as distillation and synthetic data generation, and the generalizability of the approach across different architectures and datasets were questioned. The reviewers' feedback suggests that a deeper analysis of the contribution of each component to the overall performance improvement would strengthen the paper.

---

### Decision · Program_Chairs · 2025-01-22

Reject